# The Relationship between In Vitro and In Vivo Starch Digestion Kinetics of Breads Varying in Dietary Fibre

**DOI:** 10.3390/foods9091337

**Published:** 2020-09-22

**Authors:** Patricia Rojas-Bonzi, Cecilie Toft Vangsøe, Kirstine Lykke Nielsen, Helle Nygaard Lærke, Mette Skou Hedemann, Knud Erik Bach Knudsen

**Affiliations:** Department of Animal Science, Aarhus University, DK-8830 Tjele, Denmark; projasbonzi@gmail.com (P.R.-B.); cecilie.vangsoe@anis.au.dk (C.T.V.); klyn@forens.au.dk (K.L.N.); Hellen.laerke@anis.au.dk (H.N.L.); mette.hedemann@anis.au.dk (M.S.H.)

**Keywords:** in vitro, in vivo, digestion, starch, dietary fibre

## Abstract

The relationship between in vitro and in vivo starch digestion kinetics was studied in portal vein catheterised pigs fed breads varying in dietary fibre (DF) content and composition. The breads were a low DF white wheat bread, two high DF whole grain rye breads without and with whole kernels and two experimental breads with added arabinoxylan or oat β-glucan concentrates, respectively. In vitro, samples were collected at 0, 5, 10, 15, 30, 60, 120 and 180 min and the cumulative hydrolysis curve for starch was modelled, whereas the in vivo cumulative absorption models for starch were based on samples taken every 15 min up to 60 min and then every 30 min up to 240 min. The starch hydrolysis rate in vitro (0.07 to 0.16%/min) was far higher than the rate of glucose appearance in vivo (0.017 to 0.023% absorbed starch/min). However, the ranking of the breads was the same in vitro and in vivo and there was a strong relationship between the kinetic parameters.

## 1. Introduction

Bread is a stable carbohydrate rich food consumed in various forms around the world (white wheat bread, brown bread, whole grain breads, bran enriched breads, etc.). In Denmark, bread accounts for ~17% of dietary energy intake [1] and along with other cereal products, for ~60% of the intake of dietary fibre (DF) [2]. It is known that the glycaemic response expressed in the glycaemic index (GI) varies widely depending on the type of foods. GI is a system for the exchange of carbohydrates between foods and is defined as the area under the blood glucose curve following the ingestion of a test food, expressed as a percentage of the corresponding area following an equivalent load of a reference carbohydrate, either glucose or white wheat bread [3]. In an extensive review of Atkinson et al. [4] the GI of breads varied from 34 for barley bread with 80% scalded intact kernels to 87–89 for white flour breads from different countries. The variability reflects different rates of starch digestion, where the porous structure of white wheat bread makes it easily destructed in the mouth, stomach and small intestine [5], leading to the high glycaemic response. On the other hand, starch from bread containing intact kernels is digested more slowly due to the intact botanical structure that protects encapsulated starch in the kernel against hydrolysis by enzymes in the gastrointestinal tract [6].

The presence of DF in cereals, either occurring naturally or added as concentrates, can influence the glycaemic response due to its physicochemical properties in the gastrointestinal tract. Soluble DF may form viscous solutions or gels in the stomach that delay gastric emptying and physically inhibit the rate of digestion and absorption of starch in the small intestine, (see review of Geman et al. [7]) thereby causing lower postprandial glucose and insulin responses [8,9]. The main soluble DF in oats and barley are (1→3)(1→4)-β-d-glucan (referred to as β-glucan), which has been approved by the Food and Drug Administration in the US [10] and the European Food Safety Authorities in the European Union [11] for health claims. In rye and wheat, arabinoxylan is the main soluble DF component and concentrates of arabinoxylan have been shown to reduce both glucose and insulin responses [8,12], similar to that of β-glucan [9,13].

The glycaemic response is related to the rate of starch digestion in the intestine and the GI values of cereal products have been positively and negatively related to rapidly available glucose and slowly available glucose, respectively, determined in vitro [14,15]. In the present study, we used a modification of the in vitro model for the classification of nutritionally important starch fractions [16] which has been applied to model the portal appearance of glucose in pigs [17]. In the current study, the in vitro procedure [17] was further modified by sampling more frequently (0, 5, 10, 15, 30 and 60 min compared to 0, 20, 40 and 60 min) the first 60 min and restricted the total length of incubation to 180 min rather than 360 min. The main objective of this investigation was to study the in vitro digestion kinetics of breads varying in DF content and composition. In order to evaluate the reliability of the in vitro assay as a predictor of in vivo events, in vivo data from a study with portal vein catheterised pigs [18] fed the actual breads were used.

## 2. Materials and Methods

### 2.1. Breads

The breads studied were kept frozen from a previous in vivo kinetic study with portal vein catheterised pigs [18]. They consisted of 5 breads—low DF white wheat bread (WWB), two high DF whole grain breads provided without (WRB) and with whole kernels (WRBK) and two experimental high DF breads based on wheat flour with arabinoxylan concentrate (AXB) or oat β-glucan (BGB). WWB, WRB and WRBK were commercial breads whereas AXB and BGB were the experimental breads prepared specifically for the project; for more details of the breads, see Christensen et al. [18]. The chemical composition of the breads is shown in Table 1.

### 2.2. Chemical Analysis

The breads were freeze-dried and ground to a particle size less than 0.5 mm to make the samples homogenous for chemical analyses. All analyses were performed in duplicate. Dry matter (DM) was determined by drying to a constant weight at 103 °C for 20 h, ash was analysed by the AOAC method (923.03; AOAC) [19], protein (N × 6.25) by element analysis according to the DUMAS method [20], and fat according to the Stoldt procedure [21]. β-glucan was analysed by the enzymatic-colorimetric method of McCleary and Glennie-Holmes [22], the Klason lignin as described by Theander and Åman [23], and the low molecular weight sugars as described by Kasprzak et al. [24]. Starch and non-starch polysaccharides (NSP) were analysed essentially as described by Bach Knudsen [25]; 2 M H_2_SO_4_ for 1 h was used instead of 1 M H_2_SO_4_ for 2 h for the NSP analysis. The content of non-digestible carbohydrates (NDC) was determined by direct acid hydrolysis without starch removal and alcohol precipitation and NDC calculated by the subtraction of the starch content. The content of low molecular weight ((LMW) NDC) was calculated as
LMW NDC = total carbohydrates − NSP − starch(1)

The content of resistant starch (RS) in the breads was calculated as
RS = NSP_glucose_ − (cellulose + β-glucan)(2)

### 2.3. In Vitro Starch Digestion

A modification of the in vitro starch digestion assay of Vangsøe et al. [17] and Englyst et al. [16] was used (Figure 1). Sampling was performed at 0, 5, 10, 15, 30, 60,120 and 180 min. The idea of sampling every 5 min at the beginning was to better understand the kinetics of the digestion in the early phase of the digestion. The total incubation time was chosen to be 180 min, to ensure a full digestion of the breads.

All the reactions and sample preparations were performed in Greiner 50 mL centrifuge tubes (In vitro, Fredensborg, Denmark). Before the analysis, the frozen beads (stored at −20 °C) were partially defrosted and ground (Moulinex Coffee grinder A848, Moulinex, France) for 20 s at the same speed and constant rhythm. The in vitro analyses were performed in triplicates by weighing 800 mg breads (245–360 mg of starch) into the centrifuge tubes and 50 mg guar gum (Sigma G-4129, Sigma-Aldrich, Brøndby, Denmark) to ensure similar viscosity in all the samples. Potato starch (Merck CAT No 1252, Hellerup, Denmark) was used as a reference. Glucose standards (Merck Cat No.8337, Merck Hellerup, Denmark, for 0, 10 and 30 g/Glucose/L), dissolved in 0.1 M sodium acetate buffer (Merck Cat No. 6268, Hellerup, Denmark) with 4 mM CaCl_2_ (Merck Cat No 2382, Hellerup, Denmark) were prepared in duplicates, except the blank standard which was prepared in singular.

All samples received 5 glass balls to enhance agitation and provide grinding action. Initially, the samples were incubated horizontally in a 10 mL Pepsin solution (Sigma P-7000, Brøndby, Denmark, with 50 mg of Pepsin dissolved on 10 mL 0.05 M HCl) for 30 min in a water bath at 37 °C to mimic gastric digestion. After that, 10 mL of 0.25 M sodium acetate buffer was added to neutralize the acidity, and the samples were incubated for 10 min under the same conditions. In an arranged scheme for precise timing in the collection points, an enzyme mix containing Pancreatin (Sigma P 7545, Brøndby, Denmark), amyloglucosidase (EC 3.2.1.3. Megazyme, Wicklow, Ireland, 3260 U/mL) and Invertase (EC 3.2.1.26, Megazyme, Wicklow, Ireland, 200 U/mL) was added as described by Kasprzak et al. [26]. Incubation was carried in a shaking water bath at 160 strokes/min with a stroke length of 35 mm to mimic peristaltic and segmenting movements, at 37 °C. At each sampling point, a 0.5 mL aliquot was taken out and transferred to a centrifuge tube with 35 mL of 66% ethanol to stop enzymatic digestion. Once the incubation period was over after the last sampling, samples were removed from the water bath to proceed to total starch determination. Samples were first mixed and incubated in a boiling water bath for 30 min to achieve gelatinization, then to an ice-water bath (≤5 °C) for 20 min. After cooling, 10 mL of 7 M potassium hydroxide (Merck Cat No.5000, Hellerup, Denmark) was added and the samples were again placed horizontally on an ice-water bath for 30 min to disperse any retrograded starch. After that, 1 mL of each sample was transferred to a tube containing 10 mL of 0.5 M acetic acid with 4 mM CaCl_2_. To hydrolyse the starch to glucose, 200 µL of amyloglucosidase (EC 3.2.1.3, Megazyme, Wicklow, Ireland, 408 U/mL) diluted in 5 mL of water was added to each sample and incubated in a water bath at 70 °C for 30 min. After the 30 min, the samples were placed in boiling water for 10 min, diluted in 35 mL of water and centrifuged at 2980× *g* for 10 min. The timed aliquots in the 66% ethanol solution were centrifuged at 2980× *g* for 10 min before glucose determination using a glucose oxidase kit (GODPOD, K-GLUC, Megazyme, Wicklow, Ireland) after adding 260 µL of GOD-POD to 25 µL of the sample in a microplate and incubated for 20 min in a 40 °C oven before measuring in a plate-reader (Plate Count Reader, Bio-tek EL808i, Bad Friedrichshall, Germany) at 510 nm.

The content of free glucose was determined separately, by weighting duplicated samples of 500 mg into a centrifuge tube, adding 25 mL of acetate buffer (0.1 M at pH 5.0) and incubating for 60 min in a water bath at 65 °C. After cooling to room temperature, the samples were centrifuged at 2980× *g* for 10 min and glucose was determined in a coupled reaction with NADP^+^ as described by Larsson and Bengtsson [27].

### 2.4. Reference In Vivo Data

The in vivo data were obtained from the study of Christensen et al. [18] where the glycaemic effects of breads high in DF were compared to a low-DF white wheat bread (WWB) in porto-arterial catheterised pigs. In brief, the pigs were fed each of the five experimental breads on separate days in a randomized 5 × 6 incomplete crossover design with washout periods in between to have 6 observations per bread. On the sampling days (Monday and Thursday) the morning meal was replaced with a pulse dose of one of the breads portioned to give approx. 200 g of available carbohydrate to be eaten within 15 min. For further details, see the protocol described by Christensen et al. [18].

### 2.5. Calculations and Statistical Analysis

The proportion of hydrolysed starch (%) over time in vitro was obtained from the released glucose multiplied by a factor of 0.9 to convert to anhydro sugars:(3)$$\mathrm{Hydrolysed}\;\mathrm{starch}=\frac{\left(C_{t}-C_{0}\right)\times 0.9}{\mathrm{TS}}\;\times 100$$
where C_t_ is the glucose released at time *t*, C_0_ is the glucose released at time 0, and TS is total starch. The hydrolysis of starch in the time frames 0–5 min, 5–10 min and 10–15 min were calculated by taking the cumulative starch hydrolysed at the end of the periods, at 5, 10 and 15 min, and subtracting the cumulated starch at the start, 0, 5 and 10 min, respectively.

The absorbed starch in vivo was calculated based on absorbed glucose to the portal vein, as described by [18] and converted to starch by the factor 0.9:(4)$$\mathrm{Absorbed}\;\mathrm{starch}\;=\;\frac{\left(\mathrm{Cp}-\mathrm{Ca}\right)\times F\left(\mathrm{dt}\right)\times 0.9}{\mathrm{TS}}\;\times \;100$$
(5)$$Q=\;\overset{t240}{\underset{0}{\sum }}q$$
where q is the amount of starch absorbed within the time period dt, C_p_ is the concentration of glucose in the portal vein, C_a_ is the concentration of glucose in the mesenteric artery, F is the blood flow in the portal vein at time period dt, Q is the cumulative amount of starch absorbed from *t*_0_ to *t*_300_ and TS is the total starch.

The data from the in vitro studies were analysed by a one-way analysis of variance model for each time point of the hydrolysis curve and for the generated variable:Y_ij_ = μ + α_i_ + ε_ij_(6)
where μ is the overall mean, α_i_ is the effect of diet (i.e., WWB, WRB, WRBK, AXB, BGB) and ε_ij_ describes the random error. Levels of significance were reported as being significant when *p* < 0.05.

The data from the in vivo study were analysed by a two-way analysis of variance model for each time point of the hydrolysis curve and for the generated variable:Y_ijk_ = μ + α_i_ + β_j_ + ε_ijk_(7)
where μ is the overall mean, α_i_ is the effect of diet (i.e., WWB, WRB, WRBK, AXB, BGB), β_j_ the effect of the pig and ε_ijk_ describes the random error. Levels of significance were reported as being significant when *p* < 0.05.

The relationship between the cumulated hydrolysis of starch in vitro and time was described by a mechanistic growth model:Y = α (1 − βe^−cT^)(8)
where Y is the dependent variable (cumulated starch hydrolysis at time T), α is the asymptotic value, c the growth rate, β the scale and T is the time.

The relationship between the cumulated hydrolysis of starch in vivo and time was described by a Gomperts 3P model:(9)$$Y={\alpha e}^{-e^{-\beta \left(T-c\right)}}$$
where y is the dependent variable (cumulated starch absorption at time T), α the asymptotic value, β the growth rate, c the inflection point, and T is time.

The relationship between the *k* values obtained in vitro and in vivo was analysed by a linear regression model:Y_ij_ = μ + β_i_ × X+ ε_ij_(10)
where μ is the intercept, β_i_ is the slope and ε_ij_ describes the random error.

All statistical analyses were carried out using JMP^®^, Version 13.2, (SAS Institute Inc., Cary, NC, USA).

## 3. Results

### 3.1. Dietary Composition

The WWB bread had the highest total starch content (711 g/kg DM), whereas the content was lower in all high DF breads (514–612 g/kg DM) (Table 1). The breads also contained 23–58 g/kg of sugars. Total DF was 77 g/kg DM in the WWB and 199–220 g/kg DM in the high DF breads, which also had higher contents of soluble DF (50–86 g/kg DM) than the low DF bread. The type of total and soluble DF, however, varied greatly between the high DF breads with BGB being high in total and soluble β-glucan and WRB, WRBK and AXB being high in total and soluble arabinoxylan.

### 3.2. In Vitro Starch Digestion

The content of free glucose at time *T* = 0 was significantly different (*p* < 0.05) between breads, being highest for AXB, followed by WWB, WRB and lowest for WRBK and BGB; the last two breads were not significantly different (Table 2). In vitro starch hydrolysis was described by a mechanistic growth model (*R*^2^ > 0.93) following first order kinetics. The rate of hydrolysis differed between the breads up to 30 min with the highest hydrolysis rate for WWB and the lowest for BGB, and with the other breads in between (Figure 2); for details on significance, see Appendix A.

The highest hydrolysis rate was between 0 and 5 min and with decreasing rates thereafter. In this period, the highest hydrolysis rate was found for WWB (13.9% starch/min), followed by WRB (10.4% starch/min), WRBK (8.7% starch/min) and finally AXB and BGB (7.4–8.5% starch/min) (Figure 3). From 5 to 10 min there was a marked drop in the hydrolysis rate particularly for WWB (0.18% starch/min) whereas RBK in this period had the highest value (3.9% starch/min) and with AXB, WRB and BGB in between (2.4–3.6% starch/min). The picture was almost the same although with different ranking from 10 to 15 min with the lowest value for WWB (0.4% starch/min) but with the highest value for AXB (2.2% starch/min) and with the other diets in between (1.2–1.6% starch/min). From 15 min and onward the hydrolysis rate was generally low (<1% starch/min) and it was only at 60 min that BGB had a higher value than the other breads (Appendix A). The hydrolysis plateaus (89–94%) were reached at 60 min and although numerically lower for WRBK (85.1%) and the other breads the difference was not significant.

### 3.3. Comparison to In Vivo Results

The cumulative portal glucose appearance expressed in units of hydrolysed starch per 100 g of dry starch is shown in Figure 4. Data points were described by a Gompertz 3P model, which gave a good fit to the data (*R^2^* ˃ 0.98). The cumulative starch hydrolysis after 15 min was highest for WWB and the lowest was from WRB and WRBK, with AXB and BGB in between (*p* < 0.05). At 30 min and onwards, there was no significant difference between the hydrolysis rates between the breads but the WWB continued to give the highest numeric values and WRBK and WRB the lowest ones. The cumulated starch hydrolysis approached the asymptotic values (57–72%) at approximately 240 min for all breads and with no difference between the breads.

The *k* values in vitro varied from 0.0744 to 0.1595% hydrolysis/min with significant differences between WWB and BGB (Table 2). The *k* values in vivo showed no significant difference between breads but there was a trend for a higher *k* value for WWB than the other breads. The correlation between the in vitro and in vivo *k* values in this study was *r* = 0,78 (*p* = 0.122) and when the data were combined with the data from the previous study of Vangsøe et al. [17] the correlation between the *k* values was *r* = 0.71 (*p* < 0.05) (Figure 5).

## 4. Discussion

The current study investigated the in vitro starch digestion kinetics of breads varying in total DF and the proportion between soluble and insoluble DF. Compared to the previous study of Vangsøe et al. [17], samples were taken more frequently in the first 30 min of in vitro digestion, which enables a better modelling of the digestion kinetics, particularly in the early phase of the in vitro digestion. A common feature of white wheat flour breads is its porous physical structure which makes it easily destructible [5] as seen by the very high hydrolysis rate of WWB within the first 5 min of in vitro digestion. In this way, the present study corroborates the study of Péronnet et al. [15] when substituting extruded cereals high in rapidly digestible starch with biscuits with a high content of slowly digestible starch. The presence of DF, naturally in the cell walls (WRB, WRBK) or added (AXB, BGB), delayed the in vitro digestion by decreasing the hydrolysis rate in the first 5 min but increasing it in the following 10 min relative to WWB. The strongest effect was seen for BGB, presumably because the added high *M_W_* β-glucan [28] resulted in the higher viscosity of BGB compared to the other breads [24]. Due to the linear structure, barley and oat β-glucans are very sensitive to depolymerisation, both during the technological process (baking) but also during passage of stomach and small intestine which reduces their viscosity in vivo [28]. Arabinoxylan concentrate with a more branched structure than β-glucan [29,30] also reduced the rate of in vitro digestion within the first 5 min but in contrast to β-glucan, arabinoxylan is less sensitive to modification during the passage of the stomach and small intestine. Furthermore, while the viscosity of AXB is lower than that of BGB, the viscosity of the digesta after consuming AXB at ileum is higher than that of BGB [28]. In vivo, the consequence of that is a stronger effect of AXB than BGB on the glucose flux in pigs [18] and glucose metabolism in Zucker Diabetic Fatty rats [31]. In human subjects with metabolic syndrome, however, BGB appears more efficient than AXB in influencing the glycaemic response [32], possibly because β-glucan is less modified in the upper guts of humans than that of most animals.

The WRB also resulted in a lower rate of hydrolysis the first 5 min of incubation, which corroborates with Juntunen et al. [33], who found that the hydrolysis rate of rye endosperm bread, traditional rye bread and high DF rye bread all were lower than that of the reference white wheat bread. The reason for the difference between the wheat and rye types of breads in this and the study of Juntunen et al. [33] is most likely the bread processing that gives white wheat bread a more porous structure [5] compared to the more compact structure of rye bread [33].

The inclusion of intact kernels into breads have been shown as an efficient way of regulating the rate of starch hydrolysis because the insoluble fibrous network surrounds starch, and thereby forms a physical barrier to α-amylases and limits the degree of starch gelatinization [6,34]. In the current study, the starch hydrolysis rate during the first 5 min was lower than that of WRBK, WRB and WWB as was the in vivo glucose flux 15 min post feeding [18]. In a study where the same batches of breads were fed to human subjects with metabolic syndrome, the postprandial glucose response and incremental area under the curve were also lower when consuming the WRBK than the WWB bread [32]. Our in vitro and in vivo results [18,32] are, thus, in concert with other studies, although the effect of including intact barley kernels appears to be stronger than found with rye [6].

The in vivo cumulated absorption of glucose found in this and other studies [18,35,36] are slower than the in vitro hydrolysis because food components such as fat, protein and soluble DF affect gastric emptying [37,38] through the release of cholecystokinin hormones [39]. The viscous nature of soluble DF may further increase the viscosity of the digesta, especially the thickness of the unstirred layer at the intestinal mucosa surface, which limits the diffusion and delays the absorption of glucose through the epithelial cells [40,41]. These factors, rather than the absorption rate to the portal vein, appear to be the main reason for the slower glucose absorption in vivo than in vitro, as glucose from free dietary glucose is present in the portal vein within minutes of ingestion and with a peak level after 30 min, whereas peak level is not reached until after 120 min when glucose is provided as raw maize starch [42]. In the studies of Rerat et al. [43], it was also concluded that it was not the α-amylase activity per se that limited glucose absorption but the starch structure, as can also be noticed when comparing the in vitro k-values of the breads in the current study with the in vitro *k* values of the raw diets in the study of Vangsøe et al. [17]; the higher hydrolysis rate of the breads most likely reflects the porous structure and gelatinization of the starch in the breads, making their digestion more rapid than that of the raw ingredients [5]. In spite of these differences between the in vitro and in vivo models, the strong relationship between the *k* values in vitro and in vivo obtained not only with the breads of the current study but also with the raw diets of the study of Vangsøe et al. [17] is encouraging.

For all diets, the in vivo starch digestion displayed lower digestion asymptotic values than in vitro and slightly lower than the values reported in the study of Vangsøe et al. [17]. The lower values found in vivo than in vitro in both studies were without any doubt caused by glucose utilization by the intestinal epithelium as also reported in other studies [36,44].

## 5. Conclusions

The in vitro method ranked the breads in a similar manner as in vivo but the rate of glucose release was much faster in vitro than in vivo in portal vein catheterised pigs. Although the in vivo model is more complex than the in vitro model used, it is encouraging that there is a strong correlation between the *k* values obtained in vitro and in vivo, not only with breads in this study but also when including data from a previous study, indicating that the in vitro model can provide valuable information about the digestion rate in the gastrointestinal tract of value not only in a pig model but rather in general.

## Figures and Tables

**Figure 1 foods-09-01337-f001:**
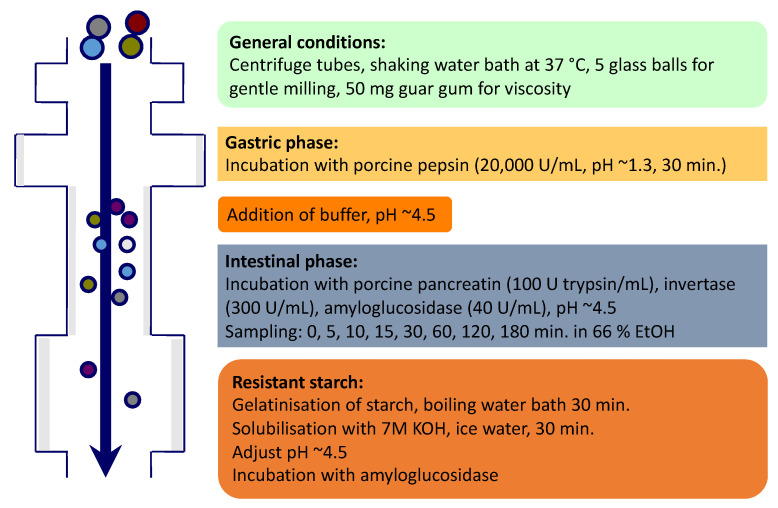
Schematic presentation of the in vitro digestion model, modified after Vangsøe et al. [17].

**Figure 2 foods-09-01337-f002:**
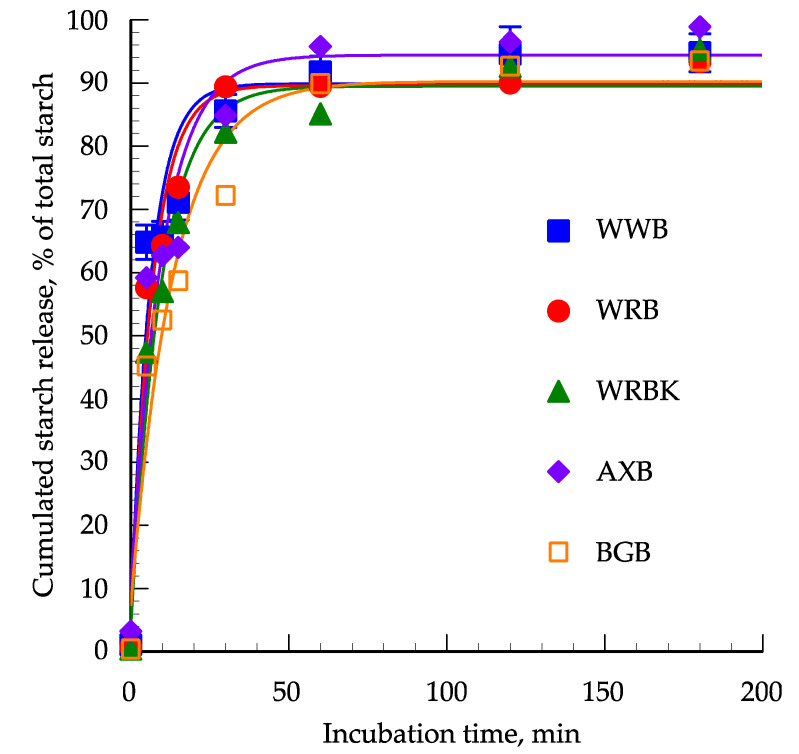
In vitro cumulative starch hydrolysis (% of total dry starch) of the white wheat bread (WWB), whole grain rye bread (WRB), whole grain rye bread with kernels (WRBK), arabinoxylan bread (AXB) and the beta-glucan bread (BGB). Values are the means ± SEM (*n* = 3).

**Figure 3 foods-09-01337-f003:**
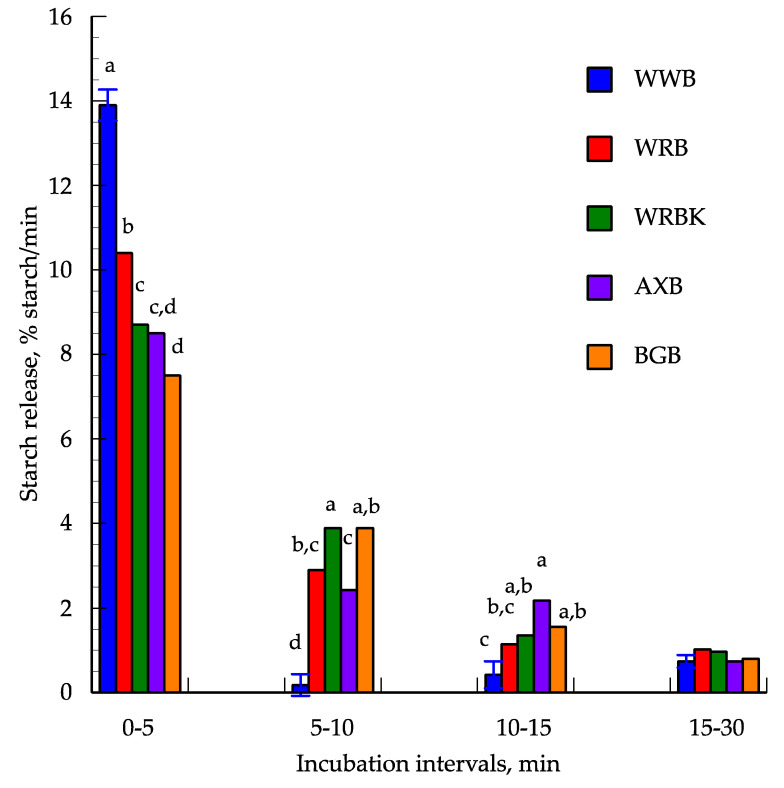
In vitro starch hydrolysis (% of total starch/min) in different time intervals of the white wheat bread (WWB), whole grain rye bread (WRB), whole grain rye bread with kernels (WRBK), arabinoxylan bread (AXB) and the beta-glucan bread (BGB). Values are the means ± SEM (*n* = 3); ^a, b, c, d^ mean values with different superscript letters were significantly different.

**Figure 4 foods-09-01337-f004:**
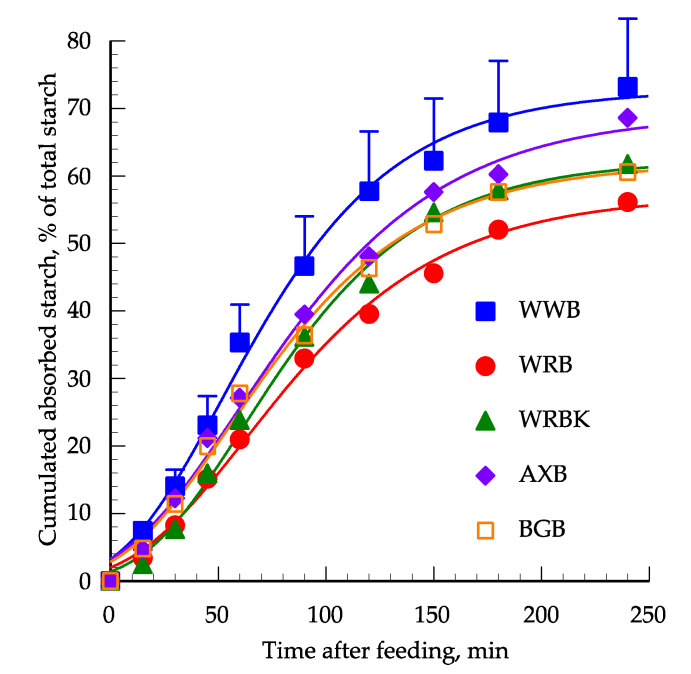
In vivo cumulative portal glucose appearance (% starch absorbed of ingested starch) of white wheat bread (WWB), whole grain rye bread (WRB), whole grain rye bread with kernels (WRBK), arabinoxylan bread (AXB) and beta-glucan bread (BGB). Values are the means ±SEM (n = 6).

**Figure 5 foods-09-01337-f005:**
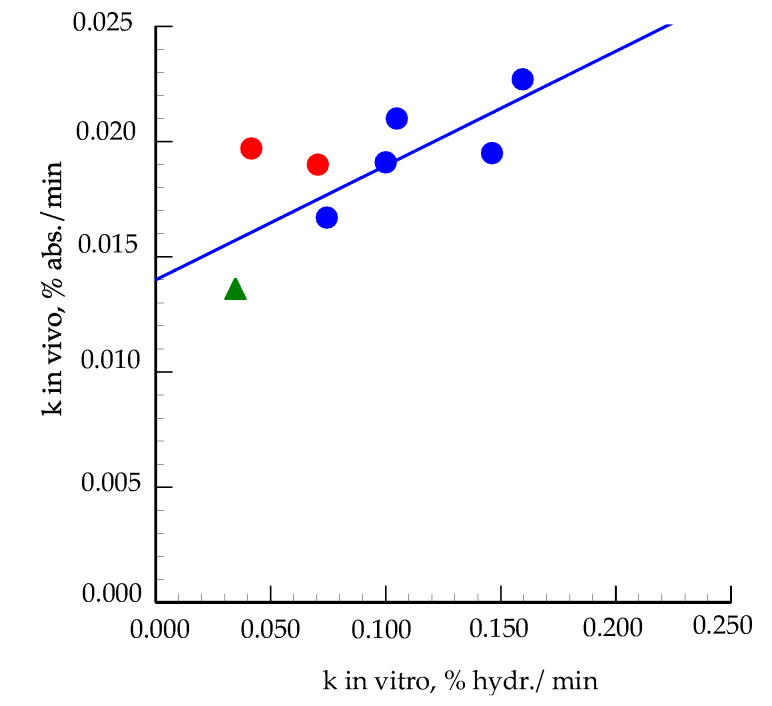
Relationship between in vitro and in vivo *k* values of the investigated breads of the current study (J) and of Western style diet (J), arabinoxylan rich diet (J) and resistant starch rich diet (H) in the study of Vangsøe et al. [17]. The relationship could be expressed with: *k*_in vivo_ = 0.014 + 0.043 × *k*_in vitro_, *R*^2^ = 0.504.

**Table 1 foods-09-01337-t001:** Chemical composition (g/kg dry matter (DM)) of the experimental test breads, see Christensen et al. [18] for details.

Chemical Composition	WWB	WRB	WRBK	AXB	BGB
Dry matter (g/kg as is)	634	520	543	708	615
Total starch	711	588	608	514	612
Total sugars	47	32	23	39	58
Dietary fibre					
LMW non-digestible carbohydrates	30	52	53	81	10
Resistant starch	4	9	14	7	18
Total NSP (soluble NSP)	35 (17)	134 (53)	139 (50)	116 (86)	163 (54)
Cellulose	6	19	18	6	53
β-Glucan (soluble β-glucan)	3 (2)	21 (7)	19 (4)	3 (2)	52 (40)
AX (soluble AX)	17 (13)	76 (36)	77 (37)	78 (66)	32 (9)
Total non-digestible carbohydrates ^a^	69	195	206	204	190
Klason lignin	8	14	14	8	9
Total dietary fibre ^b^	77	209	220	212	199

WWB, white wheat bread; WRB, whole grain rye bread; WRBK, whole grain rye bread with kernels, AXB, wheat bread with arabinoxylan concentrate; BGB, wheat bread with β-glucan concentrate; ^a^ Calculated as: total non-starch polysaccharides (NSP) + low-molecular weight (LMW) non-digestible carbohydrates + resistant starch; ^b^ Calculated as: Total NSP + LMW non-digestible carbohydrates + resistant starch + lignin.

**Table 2 foods-09-01337-t002:** In vitro and in vivo data of the digestion rate (*k*), asymptote and inflection point of breads varying in dietary fibre.

	Breads						
	WWB	WRB	WRBK	AXB	BGB	SEM	*p*-Value
In vitro							
*k*, % hydr./min	0.1595 ^a^	0.1462 ^a, b^	0.1048 ^a, b^	0.1000 ^a, b^	0.0744 ^b^	0.02	0.06
Asymptote,%	89.8	89.6	89.5	94.4	90.2	3.1	0.75
In vivo							
*k*, % absorption/min	0.0227 ^a^	0.0195 ^a^	0.021	0.0191^a^	0.0167 ^a^	0.023	0.085
Inflection point	55.4	66.0	66.0	60.6	59.8	7.3	0.82
Asymptote,%	71.8	56.8	62.6	68.5	72.2	10.4	0.85
*k*_in vitro_/*k*_in vivo_	7.02	7.50	4.99	5.23	4.45		

WWB, white wheat bread; WRB, whole grain rye bread; WRBK, whole grain rye bread with kernels, AXB, wheat bread with arabinoxylan concentrate; BGB, wheat bread with β-glucan concentrate. ^a, b^ mean values with different superscript letters were significantly different.

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
