# Peer review of "The Relationship between In Vitro and In Vivo Starch Digestion Kinetics of Breads Varying in Dietary Fibre"

_foods, 2020, doi:10.3390/foods9091337_

Round 1
Reviewer 1 Report
This article describes an in vitro method to assess the the kinetics of starch digestion and its compared for its validity with in vivo assessment based on pigs presented in an earlier report.
The study is interesting and well presented. I do not see any major problem with the way the article is written.
But, I have major concern with the experimental design.
The authors have used different types of breads which have significantly different composition and nutrient value. So, the starting material which has been used to perform this study has variation. Ideally for digestion the sample should have been normalized in terms of quantity to get standardized amount of starch with all groups. To make the experimental design clear, authors should expand the portion presented in section 2.3.
In vivo condition, after the digestion, substances have to pass thru the gut lining where the cells update nutrients for utilization and storage, only the rest in transferred to the circulation. So, there is a physical barrier, which is the reason for faster glucose release in in vitro model than the in vivo condition. This will be more predominant with sugars. This aspect of absorption is not discussed in this article.
Also, digestion pattern and efficiency is different with animals. Pigs are generally considered to be having higher metabolic rate. In such condition, comparing this in vitro model with the in vivo studies in pigs should be restricted only to pigs or at least this needs to be mentioned or properly discussed in the article. This would make the reader understand what to exactly expect from this article.
Author Response
The study is interesting and well presented. I do not see any major problem with the way the article is written.
Thank you very much for these kind words.
But, I have major concern with the experimental design.
we have tried to address the criticism in the manuscript and below.
The authors have used different types of breads which have significantly different composition and nutrient value. So, the starting material which has been used to perform this study has variation. Ideally for digestion the sample should have been normalized in terms of quantity to get standardized amount of starch with all groups. To make the experimental design clear, authors should expand the portion presented in section 2.3.
I have added information about the amount of starch from each breads (L91). A general feature of the original models our modification is build on is that the enzymes are present in excess (Englyst, H.N.; Kingman, S.M.; Cummings, J.H. Classification and measurement of nutritionally important starch fractions. Eur. J. Clin. Nutr. 1992, 46, S33-50) as is also the case under in vivo conditions (Rérat, A.A.; Vaissade, P.; Vaugelade, P. Absorption kinetics of some carbohydrates in conscious pigs. 2. Quantitative aspects. Br J Nutr 1984, 51, 517-529).
In vivo condition, after the digestion, substances have to pass thru the gut lining where the cells update nutrients for utilization and storage, only the rest in transferred to the circulation. So, there is a physical barrier, which is the reason for faster glucose release in in vitro model than the in vivo condition. This will be more predominant with sugars. This aspect of absorption is not discussed in this article.
We would like to mention that in the pig model used, blood is collected simultaneously from the portal vein and mesentery artery so the only tissue the blood is passing is the intestinal wall and as seen of the works of Rerat et al (Rérat, A.A.; Vaissade, P.; Vaugelade, P. Absorption kinetics of some carbohydrates in conscious pigs. 1. Qualitative aspects. Br J Nutr 1984, 51, 505-515) that glucose is present in the portal vein more or less immediately after being ingested. This has been expanded upon in L285-291.
Also, digestion pattern and efficiency is different with animals. Pigs are generally considered to be having higher metabolic rate. In such condition, comparing this in vitro model with the in vivo studies in pigs should be restricted only to pigs or at least this needs to be mentioned or properly discussed in the article. This would make the reader understand what to exactly expect from this article.
We agree on this point and has tried to elaborate a bit on that in the conclusion, see L303-309.

Reviewer 2 Report
Manuscript ID: Foods-931704
Title: In vitro starch digestion kinetics of breads varying in dietary fibre content and composition, compared with in vivo portal appearance of glucose in pigs.
The authors described the in vitro digestion kinetics of five types of breads varying in Dietary Fiber content and composition. In addition, the relationship between in vitro and in vivo starch digestion kinetics was studied in portal vein catheterised pigs, fed with the different kind of breads.
The work is well described, clear and expanded in every section. I would suggest to the authors only a few changes.
Decision: Minor revision
Abstract and in all the manuscript: replace “min.” with “min”
Lines 49-56: This part is not clear: what in the here applied in vitro model, has been modified (times, liquids composition, enzymes concentration, etc.)? In my opinion, since the aims encourages the reader to continue, it would be useful to better explain these sentences.
Lines 88-90: The Authors referred to the bread samples freeze-dried (lines 67-68)? Why is a further grinding done, since in the previous step the 0.5 mm particles were obtained? It is not clear, please specify ...
Figure 2. : I suggest to the Authors to change the horizontal axis scale, because as it is reported it is not easy to identify the minutes. In fact the internal sign represents 12 min 30 sec?
Figure 3. : Please, dwindle the dimension of the superscript letters, becouse the reader might get confused.
Author Response
The work is well described, clear and expanded in every section. I would suggest to the authors only a few changes.
Thanks for these nice words.
Abstract and in all the manuscript: replace “min.” with “min”
This has been changed throughout the manuscript.
Lines 49-56: This part is not clear: what in the here applied in vitro model, has been modified (times, liquids composition, enzymes concentration, etc.)? In my opinion, since the aims encourages the reader to continue, it would be useful to better explain these sentences.
This has been better explained, see new L51-53.
Lines 88-90: The Authors referred to the bread samples freeze-dried (lines 67-68)? Why is a further grinding done, since in the previous step the 0.5 mm particles were obtained? It is not clear, please specify ...
The grinding to pass 0.5 mm particle size is for the chemical analyses and to make them homogenous to sample from, see L67-68.
Figure 2. : I suggest to the Authors to change the horizontal axis scale, because as it is reported it is not easy to identify the minutes. In fact the internal sign represents 12 min 30 sec?
The figure has been changed.
Figure 3. : Please, dwindle the dimension of the superscript letters, becouse the reader might get confused.
The figure has been revised.
Round 2
Reviewer 1 Report
I am convinced with the revisions, I think the article can be accepted.